# Foliar Application of Salicylic Acid to Mitigate Water Stress in Tomato

**DOI:** 10.3390/plants11131775

**Published:** 2022-07-05

**Authors:** Eduardo Santana Aires, Andrew Kim Lopes Ferraz, Beatriz Lívero Carvalho, Fabricio Palla Teixeira, Fernando Ferrari Putti, Emanuele Possas de Souza, João Domingos Rodrigues, Elizabeth Orika Ono

**Affiliations:** 1Department of Horticulture, School of Agronomy, São Paulo State University (Unesp), Botucatu 18618-000, Brazil; andrewkim.lf@gmail.com (A.K.L.F.); beatrizlivero@outlook.com (B.L.C.); fabriciopallas@yahoo.com.br (F.P.T.); emanuele.possas@unesp.br (E.P.d.S.); 2School of Sciences and Engineering, São Paulo State University (Unesp), Tupã 17602-496, Brazil; fernando.putti@unesp.br; 3Department of Botany, Institute of Biosciences, São Paulo State University (Unesp), Botucatu 18618-000, Brazil; joao.domingos@unesp.br (J.D.R.); elizabeth.o.ono@unesp.br (E.O.O.)

**Keywords:** floral abortion, photosynthesis, plant regulation, *Solanum lycopersicum*, water deficit

## Abstract

Salicylic acid (SA) is an important plant regulator reported as a mitigator of water deficit in plants, however without a recommendation for use in field conditions. Thus, this research aims to validate the use of SA under field conditions in regions with low water availability. For that, we evaluated CO_2_ assimilation (*A*), stomatal conductance (*g_s_*), transpiration (*E*), water use efficiency (*WUE*), and carboxylation efficiency (*A/Ci*) at 15, 30, and 45 days of continuous stress water deficit, as well as the application of salicylic acid (0.0; 0.5; 1.0; 1.5; 2.0 mM) in tomato plants subjected to continuous water deficit (45 days), in two years (2019 and 2020). The water deficit reduced the *A*, *g_s_*, *E* and *A/Ci*, while the foliar application of SA increased these parameters in all evaluated times, resulting in similar or even higher values than in plants without water deficit. Water deficit caused floral abortion in tomato plants, without the application of SA, reducing the number of fruit production. In contrast, plants that received about 1.3 mM of SA increased *A* and *A/Ci* and translocated the photo-assimilates, mainly to flowers and fruits, reducing floral abortion and increasing fruit production. Thus, foliar application of SA was efficient in mitigating the deleterious effects of water deficit in tomato plants regarding the gas exchange and fruit production.

## 1. Introduction

Climate Change has become a challenge for global food security [1], as rising temperatures and reduced water availability decrease crop productivity. Future scenarios indicate that water shortages will affect 50% of agricultural land by 2050 [2]. Besides, water deficit is the biggest environmental constraint that limits field crops and vegetables [3,4].

Among other culinary vegetables, the tomato crop is sensitive to water restriction, negatively affecting growth, production, and fruit quality [5]. Due to its high commercial demand, tomato cultivation is widespread worldwide, being cultivated on more than 5.03 million hectares, both in the field and under-protected cultivation [6].

Since tomato cultivation is mostly carried out in the open field, one of the factors that most affect its development is water availability. Thus, water deficit affects plant growth and development by negatively impacting cell division and elongation generating adverse effects on plant physiology, morphology, and ecology [7]. The limitation in growth occurs mainly because it causes a reduction in photosynthesis. This reduction results from the decrease in the activity of the enzyme ribulose 1,5-bisphosphate carboxylase/oxygenase (RuBisCo), caused by the limitation of the amount of CO_2_ in the intercellular space [8,9], which occurs due to stomatal closure.

With low water availability, stomatal closure occurs as an early response [10]. Stomatal functionality is regulated by hydraulic signals [11], although, under water deficit conditions, stomata also respond to chemical or hormonal signals produced by dehydrated roots. The most important hormonal signal in this regard is abscisic acid (ABA), found in high concentrations in plants under water deficit conditions [12]. As hormonal signaling occurs for stomatal closure and cascading effect on CO_2_ assimilation by the plant, a possible alternative to mitigate the deleterious effects of water deficit in plants is the application of plant regulators. Advances in agronomical practices, traditional breeding and modern biotechnological tools have been used to prevent the yield losses due to drought stress [13]. Adapting crops to drought stress would be the most economical strategy to improve water use efficiency and crop productivity [14].

Exogenous application of phytohormones and biostimulants is known as the effective adapting methods [15,16]. Among these salicylic acid (SA) can be mentioned, which is a phenolic compound with action on plant growth, ion absorption, and substance transport. SA is considered an important plant signaling molecule for the defense response of plants, increasing plant tolerance to biotic and abiotic stresses [17,18].

SA is part of several plant processes and its role in mitigating biotic or abiotic stresses has already been reported. For example, water deficit reduced stomatal conductance, transpiration, and CO_2_ assimilation in rice plants, while SA application increased gas exchange characteristics in plants under water deficit [19]. Habibi (2012) also demonstrated that net photosynthetic rate, transpiration rate, and stomatal conductance increased in barley plants under water deficit treated with SA [20].

Positive effects of SA application were observed on the growth and biomass accumulation of *Portulaca oleracea* under water deficit due to the maintenance of photosynthetic pigments and increased CO_2_ assimilation [21]. Chavoushi et al. (2020) reported that treatment with SA in safflower (*Carthamus tinctorius* L.) under water deficit improved photosynthesis rate, anthocyanin content, and phenylalanine ammonium lyase (PAL) enzyme activity however, it did not affect the accumulation of plant dry matter [22]. Recent results have shown that the foliar application of SA in grape tomatoes acts as water deficit mitigation [5]. However, grape tomatoes are not the most cultivated worldwide and also, as experimental conditions of this study cannot be replicated at the field level, that raises the importance of validating SA doses in situations that farmers can apply.

Natural biostimulants as eco-friendly materials include any substances applied to plants to enhance nutritional efficiency, abiotic stress tolerance, and/or crop quality traits [23]. Hence, considering that the action mechanism of SA can differ between plant species, stress levels, environmental conditions and others, it is important to evaluate the effect of SA on tomato plants under water deficit in field conditions to make the production of this culinary vegetable more sustainable, as well as to suggest options for cultivation in the regions with low water availability.

Based on the research reports, our research hypothesizes that foliar application of salicylic acid can reduce the deleterious effects of water deficit on gas exchange and tomato fruit yield. To test this hypothesis, we evaluated CO_2_ assimilation, stomatal conductance, transpiration, water use efficiency, carboxylation efficiency, and productivity of tomato plants subjected to water deficit under the application of different doses of salicylic acid.

## 2. Materials and Methods

### 2.1. Plant Material and Cultivation Conditions

The experiment was carried out at the Teaching, Research and Production Farm of São Manuel, in São Manuel city, São Paulo state, belonging to the School of Agronomic Sciences, Botucatu Campus, Sao Paulo State University, UNESP (22°44′ S de, 47°34′ W).

The climate of the region is classified as humid subtropical mesothermal. Pots (15 dm³) were filled with a Red Yellow Latosol, in sand phase, collected on the farm, with the following characteristics: pH (CaCl_2_) = 4.2; MO = 9.0 g dm^−3^; P(resin) = 2.0 mg dm^−3^; H + Al = 31 mmol_c_ dm^−3^; K^+^ = 0.6 mmol dm^−3^; Ca^2+^ = 3.0 mmol_c_ dm^−3^; Mg^2+^ = 1.0 mmol_c_ dm^−3^; CTC = 36 mmol_c_ dm^−3^, and V = 12%.

Soil treatment was performed with dolomitic limestone, foundation fertilization with simple superphosphate, potassium chloride, and urea. Fertigation was carried out with calcium nitrate, potassium chloride, magnesium sulfate, and monoammonium phosphate (MAP), as described [24]. The temperature and relative humidity of the air was monitored inside the greenhouse during the experimental period (Figure 1).

### 2.2. Treatments

Two production cycles were conducted between July and December 2019 and 2020. The experiment was carried out in randomized blocks with six treatments: application of five doses of SA (0.0; 0.5; 1.0; 1.5; 2.0 mM) in plants under water deficit with 70% of ETc (n = 4) and the control with 100% crop evapotranspiration (ETc) without application of salicylic acid (SA), four blocks were used, each plot consisted of four useful plants. For irrigation management and imposition of treatments under water deficit, the water depth was defined by measuring the vessel capacity by weighing lysimetry, with irrigation applied daily to maintain this capacity, based on the plants submitted to full irrigation (100% of ETc). Water deficit started at 15 days after sowing (DAS) and ended at 60 DAS, totaling 45 days of exposure to water restriction. The period was defined in a first test trial, where plants under continuous water deficit did not withstand a period longer than 45 days of water limitation. After 60 DAS, plants were irrigated at total capacity until the harvest (120 DAS).

The seedlings were transplanted into 15 dm³ pots when they were on average 0.15 m high, with a spacing of 1.0 m between rows and 0.5 m between each plant in a row, conducted vertically and tutored with the help of bamboos. The experiments used the tomato hybrid Colossal, of the Italian type of determined growth, developed by the company Sakata Seed Sudamérica^®^. The Italian type tomato is one of the most consumed consumers in the world, and it is cultivated in open fields or a protected environment, we chose this variety because it is representative of cultivation in different regions and because it has resistance to: Tomato Yellow leaf curl begomovirus (TYLCV), tomato spotted wilt tospovirus (TSWV), *Meloidogyne incognita* (Mi), *Meloidogyne javanica* (Mj), *Fusarium oxysporum* f. sp. *lycopersici* 1, 2 e 3. It is also not drought tolerant.

### 2.3. Foliar Application

Solutions containing salicylic acid for each treatment were prepared by dissolving salicylic acid in 5 mL of absolute ethanol, topped up with distilled water and applied from 15 DAS weekly, until 60 DAS, totaling 7 applications. Applications were carried out using a manual CO_2_ pressurized sprayer, with 0.3 kgf per 31 cm^2^ with full conical nozzles. In each application, 35 mL of the treatment solution was applied per plant. The plants under water deficit and without application of SA and the control plants received an application of water and 5 mL of absolute ethanol in the same volume as applied to the plants treated with SA.

### 2.4. Gas Exchange

The photosynthetic parameters related to gas exchange were evaluated at 15, 30, and 45 days of water deficit (DWD). For this purpose, fully expanded leaves were selected in the middle third of the plant and then readings were taken between 8:00 and 11:00 am.

The CO_2_ assimilation rate (*A*, μmol CO_2_ m^−2^.s^−1^), transpiration rate (*E*, mmol water vapor ·m^−2^.s^−1^), stomatal conductance (*g_s_*, mol·m^−2^.s^−1^), and internal CO_2_ concentration in tomato leaf (*Ci*, μmolCO_2_ ·mol^−1^·air) were measured in three plants per treatment with a portable open gas exchange system (LI-6400, LICOR). The CO_2_ concentration entering the leaf cuvette (LCF chamber; 2 cm^2^, LI-COR) averaged 400 μmol.mol^−1^, as provided by the 6400-01 CO_2_ mixer (LI-COR). The photosynthetic photon flux density (PPFD) was provided by an artificial light-emitting diode (LED) light source (6400-40 LCF, LI-COR; 90% red and 10% blue spectra), which was set to provide 1000 μmol photons·m^−2^·s^−1^ in the leaf cuvette, based on the curve of light performed previously. The vapor pressure deficit (VPD) inside the leaf cuvette was 2.08 ± 0.18 kPa, which means the relative humidity in the (sample) chamber was 65.1 ± 2.3%; water use efficiency (*WUE*, μmolCO_2_ [mmol H_2_O^−1^]) determined through the relations between CO_2_ assimilation and transpiration rate; and the instant carboxylation efficiency (*A/Ci*) determined through the relation between CO_2_ assimilation rate and the internal CO_2_ concentration of in tomato leaf.

### 2.5. Productive Characteristics

The fruit harvest started at 75 DAS and ended at 120 DAS, with the fruits of each pot being separated into commercial and non-commercial according to the Tomato Classification Norms [25] and then counted, thus determining the total number (TNF) and commercial number (CNF) of fruits. Afterwards, the fruits were weighed with a scale, determining the mass of the fruits and, by the sum of the production of each harvest, the total and commercial production of fruits in kg plant^−1^ was determined.

### 2.6. Statistical Analysis

The data obtained in the 2019 and 2020 cycles were analyzed separately, without considering them as a factor and the dates of gas exchange analysis.

The data were previously submitted to the Anderson Darling homogeneity test for statistical analysis using the Minitab software. Once the normality of the data was verified, analysis of variance (F Test) and regression analysis was performed, using the R statistical software. The control treatment (no application of SA and no water deficit) was compared to the other treatments through Dunnett’s Test, adopting a significance level of 5% probability (*p* < 0.05) on Sigma Plot 11 software [26].

## 3. Results

The results indicated a significant effect of SA on tomato plants under water deficit for all variables evaluated at different times. In a preliminary test, it was observed that the maximum period of water deficit (70% ETc), under the conditions of this research, was 45 continuous days and, for this reason, the restriction imposition was 45 days at the beginning of tomato development (critical period for establishment and production of the crop). Starting from this period, we evaluated the effect of SA on gas exchange in plants at 15, 30 and 45 days of water deficit (DWD) to understand the action of SA during the stress caused by water deficit and the dose needed to mitigate the deleterious effects of this stress on tomato photosynthesis.

### Gas Exchange

Foliar application of salicylic acid positively and significantly affected all evaluated gas exchange characteristics: CO_2_ assimilation (*A*), stomatal conductance (*g_s_*), transpiration (*E*), the internal concentration of CO_2_ (*Ci*), water use efficiency (*WUE*), and carboxylation efficiency (*A/Ci*), in the regular evaluations at 15, 30, and 45 DWD, in the two consecutive years (2019 and 2020).

For *A* (Figure 2A,C,E), increments were observed up to the maximum point curve, 1.4 and 1.2 mM of SA at 15 DWD, in the two cycles, respectively. On that evaluation date, the use of SA at the mentioned doses, under water deficit, provided similar CO_2_ assimilation for plants without water restriction (Figure 2A). At 30 DWD (Figure 2C), the dose of 1.4 mM of SA was the one that provided the highest *A* for the two years of cultivation when compared to plants without water deficit. Plants that received this dose of SA showed higher CO_2_ assimilation. For the last evaluation, at 45 DWD (Figure 2E), the dose of 0.9 mM of SA favored higher *A* to the point that plants under water deficit showed similar values to plants without water deficit in 2019. In 2020, the dose of 0.5 SA mM gave higher results for *A*, reaching values higher than those of plants without water restriction.

The *g_s_*, also increased with the 1.4 mM and 1.1 mM doses of SA at 15 DWD (Figure 2B). In 2019, the application of SA at all tested doses caused stomatal conductance similar to that of plants without water restriction. In 2020, *g_s_* was higher in plants treated with SA when compared to plants without water restriction.

At 30 DWD (Figure 2D), following the response and inflection point adjustment, the doses of 2.0 mM and 1.1 mM of SA caused greater stomatal opening in plants under water deficit in 2019 and 2020 respectively. In 2019, plants that received a dose greater than 0.5 mM of SA showed higher *g_s_* when compared to plants without water restriction. In 2020, the application of SA generated greater *g_s_* in plants under water deficit when compared to those of the control group.

The doses of 1.1 and 0.6 mM of SA provided higher *g_s_* at 45 DWD (Figure 2E) in tomato plants under water deficit in 2019 and 2020, respectively. Due to the stomatal opening, CO_2_ enters the substomatic chamber. Thus, the carbon of the CO_2_ will be assimilated, and this fact can be verified by the variable *Ci* (Figure 3A,C,E), in which lower values were expected to indicate the best treatments.

Therefore, doses of 1.8 and 0.85 mM of SA in tomato plants under water deficit caused a greater reduction of *Ci*, in the years 2019 and 2020, respectively, at 15 DWD (Figure 3A). Comparing with the plants of the control treatment, in 2019, the application of SA reduced the *Ci* values at the level of plants without water restriction, while, in 2020, the control plants showed higher *Ci* when compared to the plants in which the aforementioned SA dose was applied. At 30 DWD (Figure 3C), the same dose (1.3 mM of SA) was efficient in the reduction of *Ci* in the two years studied. They also showed similar values of this variable when compared to the control plants. For the last evaluation, at 45 DWD (Figure 3E), the dose of 0.8 mM of SA promoted the greatest decrease in *Ci* values in the two studied years. The *Ci* values were similar between the control plants and the tomato plants that received this dose of SA in water restriction.

With the increase in stomatal conductance, an increase in transpiration of tomato plants treated with SA under water deficit was also observed (Figure 3B,C,F). At 15 DWD (Figure 3B), higher values of *E* were observed at doses of 1.4 and 0.9 mM of SA in the years of 2019 and 2020, respectively. In 2019, the *E* of the plants treated with the mentioned dose was similar to that of the control plants, while, in 2020, the plants that received 100% ETc presented a higher *E* than the others. At 30 DWD (Figure 3D), the dose of 0.75 mM of SA was the one that provided the highest *E* in both years. In 2019, plants treated with the aforementioned dose showed *E* superior to the control group, and in 2020, the control and SA-treated plants showed similar values.

For the evaluation at 45 DWD (Figure 3F), the dose of 0.9 mM of SA was responsible for the largest increase in *E* in the both years of experimentation. In 2019, the application of this dose caused higher values than the control treatment, and in 2020, similar values.

The *WUE* (Figure 4A,C,E), ratio between *A* and *E*, was higher when the dose of 1.61 and 2.0 mM of SA was applied in 2019 and 2020, respectively, at 15 DWD (Figure 4A). In 2019, the control plants showed a higher *WUE* when compared to the water deficit with SA application. In 2020, the *WUE* of the plants at the same doses was higher. At 30 DWD (Figure 4C), similar behavior was observed in both years: the dose of 2.0 mM of SA provided higher values of this relationship in tomato plants under water restriction, also higher than those observed in control plants. For the last assessment, at 45 DWD (Figure 4E), the 0.9 and 0.3 mM doses of SA increased the *WUE* in 2019 and 2020, respectively. In 2019, the *WUE* was similar to that of the control plants, while in 2020, it was higher.

Another important relationship is the *A/Ci*, because through this ratio, it is possible to infer the efficiency of the Rubisco enzyme. The application of SA also positively influenced the carboxylation efficiency (Figure 4B,D,F). At 15 DWD (Figure 4B), doses of 1.5 and 1.1 mM of SA provided higher *A/Ci* values in 2019 and 2020, respectively. For both years, those doses promoted an increase in this variable to values similar to those of plants without water deficit. At 30 DWD (Figure 4D), the dose of 1.3 mM of SA was the one that promoted the greatest increases in *A/Ci*, in the two years of cultivation, to the point of being superior to the control plants. At 45 DWD (Figure 4F), the doses of 0.9 and 0.3 mM of SA provided higher *A/Ci* values in 2019 and 2020, respectively. In 2019, the *A/Ci* was similar between the control plants and those that received the mentioned dose. In 2020, tomato plants treated with the dose of SA showed higher values than the control group.

Positive effects caused by SA on gas exchange of tomato plants under water deficit, especially in CO_2_ assimilation and carboxylation efficiency, were reflected in plant production. Total production (Figure 5A) reached its maximum with the dose of 1.1 mM of SA in 2019, while in 2020, the dose of 1.6 mM of SA promoted the greatest response. In 2019, the control plants showed higher production, without significantly differing from the plants treated with SA under water deficit. While in 2020, the application of the aforementioned dose increased tomato production to the level of control plants that did not undergo water restriction.

For commercial production (Figure 5B), in 2019, the dose of 0.95 mM of SA provided the highest production, as well as favoring the production of commercial fruits. It can be observed that plants subject to water deficit and application of SA presented commercial fruit production similar to those of plants that did not suffer water restriction. For the 2020 cycle, the dose of 1.3 mM of SA increased the commercial fruit production of plants under water deficit and also presented values similar to those of plants that were not under water restriction.

The application of SA was also efficient in increasing the number of fruits per plant. This is probably due to the observed reduction in floral abortion, which increases the number of total and commercial fruits. For TNF (Figure 5C), in 2019, the dose of 0.5 mM of SA was efficient in maintaining the flowers until fruit formation, which is why the plants of this treatment presented TNF similar to the control plants. In 2020, the dose of 1.1 mM of SA was efficient in maintaining the flowers, however, it did not reach the values of the control plants. The same behavior of TNF was observed for CNF (Figure 5D), in 2019. The dose of 0.5 mM of SA was the most efficient and provided a number of commercial fruits similar to that of the control plants. In 2020, the dose of 1.8 mM of SA in plants under water deficit promoted higher CNF, without, however, showing similarity to the control plants.

## 4. Discussion

Our results suggested that the foliar application of 1.3 mM of salicylic acid increased by 55% the *A*, 58% the *A/Ci* of tomato plants under water deficit, causing a reduction in floral abortion, which represents about 25% more of total and commercial fruits. With the increase of *A* and *A/Ci*, the photoassimilates may have been translocated to the fruits, resulting in a 30% increase in the commercial production of fruits in plants treated with AS under water deficit.

During their life cycle, plants are exposed to unfavorable conditions that cause stress and these can be of biotic or abiotic origin [27]. Water deficit is a major abiotic stress of the latter kind and that inhibit plant metabolism [28]. One of the first plant responses to water deficit is stomatal closure, which results in reduced photosynthesis [29]. In transgenic tomato plants, results indicated that in significant stressful situations, like water deficit or phosphorus deficiency for example, glycinebetaine increased content can inhibit the accumulation of ROS and/or act on the activation of the plasma membrane HC-ATPase which can enhance the transport of phosphorus. However, non transgeninc plants cannot count on this strategy, since this tomato plants are not able to accumulate GB under both normal and stress conditions [30,31]. In this way, there is a reduction in the parameters of *A*, *g_s_*, and *E*, with the decrease in water availability in the soil. This relationship was observed in tomato plants under water deficit without SA application. Similar behavior was recognized in ryegrass under water deficit conditions [32].

Nevertheless, the dose of 1.4 mM of SA promoted a higher *g_s_* in tomato under water deficit and, consequently, increased *A* values. At certain moments of the evaluation, the assimilation of CO_2_ and stomatal conductance of plants under water deficit and treated with SA was similar to those of plants of the control group, indicating that this plant regulator can control the stomatal closure caused by water deficit.

At 30 e 45 DWD in 2020, no differences were found in *A* and *g_s_* between control and water deficit without SA application; in fact, the water defect was imposed because the *g_s_* was low in the treatments without SA and under deficit, while the control plants were under environmental stress (high temperature, high VPD, low humidity), and the SA, in turn, was able to alleviate water deficit and environmental stress in tomato plants. The SA application in tomato plants minimized the effects of environmental stress by increasing the activity of the enzymes SOD, CAT and POD and, reducing lipid peroxidation, protecting the photosynthetic apparatus, ensuring the proper functionality of the PSII. Similarly, Sohag et al. [33] confirms that the application of SA might activate plant defensive system and helped plant to adjust the water status under drought, due to the alleviation of drought-induced over-accumulation of ROS, possibly by enhancing the activities of antioxidant enzymes. Also, SA improved *g_s_* which increased *A*, that is, greater relation of *A/Ci*; consequently, reflecting on plant height and weight accumulation in fruits [34].

Plant regulators can lead to water deficit tolerance by modifying biochemical and physiological processes, such as maintaining stomatal opening. Although a complex process, the stomatal movement has abscisic acid (ABA) as an important regulatory component to govern it in response to reduced water availability [35]. It also stimulates stomatal closure through secondary messengers, such as reactive oxygen species (ROS), nitric oxide, calcium, and protein kinases [36]. The application of SA, in turn, reduced the content of ABA and ethylene, resulting in higher stomatal conductance and photosynthesis of mustard plants under water deficit [37].

The hormonal balance is modified in plants under water deficit, promoting an increase in ethylene and ABA concentrations as an adaptation strategy to reducing water availability. Samui et al. (2020) reported that an alternate wetting and drying situation in the soil can be conducive for increased ABA concentration in xylem sap of tomato plants [38]. However, at the same time that it works to protect plants from water deficit, it can compromise photosynthesis and plant growth. As a management strategy for crops, foliar application of SA plays a key role in reducing the ethylene and ABA content [37]. Thus, it is interesting to observe that under water deficit conditions, foliar application of SA increased CO_2_ assimilation, stomatal conductance, and water use efficiency of tomato plants, possibly by minimizing the stress caused by ROS that reduced the formation of ethylene and ABA.

Similarly, to the behavior of *g_s_*, tomato plants that received a foliar application of SA of about 1.0 mM showed higher E. Transpiration is closely related to the water status of the plant, probably due to better water absorption. Thanks to the accumulation of osmotic substances, such as soluble sugars and proline [39]. The increase in this variable is also important to regulate leaf temperature and, thus, favor the action of metabolic processes such as photosynthesis. Hence, Richards et al. (2002) stated that stable production requires high transpiration, stomatal conductance, and mesophilic conductance [40].

In cultivation conditions without water deficit, plants show high transpiration due to the maintenance of CO_2_ input through the stomata, a fact observed in tomato plants without water restriction that showed high *E*. In this context, the foliar application of SA on tomato plants under water deficit also increased *E*, which maintained the leaf temperature adjusted for the functioning of the Rubisco enzyme, a fact observed with the increase of *A/Ci*, which boosted total and commercial production numbers.

The lower *A/Ci* in tomato plants under water deficit and without SA may be related to changes in any biochemical reactions or changes in thylakoid membrane composition caused by water deficit [41] since *Ci* was high in these plants. Thus, the supply of CO_2_ to Rubisco was not compromised. The reduction in carboxylation efficiency may be linked to the non-functioning of metabolic and enzymatic processes essential for *A*, which was reduced in the plants that received SA treatment. Within this context, the role of SA in increasing the *A/Ci* of plants under water deficit may be linked to the prevention of auxin oxidation [42], whose high content increases CO_2_ assimilation in the leaf. The application of SA also enhances the action of the carbonic anhydrase (CA) enzyme [43], which facilitates the diffusion of CO_2_ through the chloroplast membrane, catalyzing the hydration of dissolved CO_2_ as it enters the lower stromal alkaline environment [44]. CA also catalyzes the reversible hydration of CO_2_ and maintains a constant supply of CO_2_ for the Rubisco enzyme.

Other researches also showed that *A/Ci* was closely associated with SA application, demonstrating a positive regulatory role in increasing CO_2_ fixation in corn plants under saline stress [45].

Salicylic acid can also affect on flowering in a variety of plants, increasing the permanence of flowers in the plant and regulating flowering time. It can also act in defense of the plant, seeking reproductive development [17,46]. The exogenous application of 1.0 mM of SA increased the number of inflorescences of marigolds (*Calendula officinalis* L.) [47]. Under water deficit conditions, it was not possible to observe an inducing effect on tomato flowering. However, the foliar application of this plant regulator resulted in lower floral abortion in tomato plants, which resulted in increases in the total and commercial numbers (TNF and CNF) of fruits. Tomato plants from the control group, which grew in a field condition without water deficit, presented the expected TNF and CNF. In contrast, plants under water deficit and without application of SA showed lower values for both due to floral abortion caused by water restriction. The application of SA reduced the stress of the water deficit and minimized the fall of the flowers, increasing the accumulation of photo-assimilates in the fruits that, resulted in enhancement in the total and commercial production of fruits.

In this sense, it has been reported that SA affects improving photosynthetic capacity due to the stimulation of the Rubisco enzyme and the increase in photosynthetic pigments [48]. This statement corroborates the results of this research where the carboxylation efficiency (*A/Ci*), which shows the activity of the Rubisco enzyme, increased about 60% in plants treated with SA compared to plants without SA treatment and under water deficit. The increase in *A* and *A/Ci* resulted in boosts in total and commercial tomato fruit production, around 40 and 45%, respectively. In field conditions, without water deficit, according to other researchers, the foliar application of 0.05 mM of SA increased the production of tomato fruits [49], and the dose of 0.1 mM was efficient to increase cucumber production [50].

The effectiveness of exogenous SA application depends on the plant species, stage of development, applied concentration, application method, and environmental conditions [51,52]. Thus, for the use of SA in plants under water deficit, the application of higher doses of this plant regulator is important to mitigate the lack of water effects.

Adjusting photosynthetic capacity under water restriction is vital for plant survival. Tomato plants under water deficit presented the modulation of gas exchange as a strategy, such as the reduction of stomatal conductance (*g_s_*) and transpiration (*E*), which resulted in lower assimilation of CO_2_ (*A*). On the other hand, foliar application of SA resulted in an increase in *g_s_* and *E* and, consequently, in *A*. We also observed an effect on *A/Ci*, which increased in tomato plants treated with SA. This increase resulted in a better distribution of photo-assimilates to flowers and fruit. Thus, floral abortion was reduced, and the fruits accumulated mass (Figure 6).

According to our results, foliar application of SA is a technique capable of mitigating the deleterious effects of water deficit on gas exchange and tomato production. Hence, it can be used to manage this tomato in regions with low water availability.

## 5. Conclusions

With our results, we can conclude that tomato plants showed to be sensitive to water deficit, with a reduction in gas exchange and fruit production when only 70% of ETc was replaced. However, the foliar salicylic acid application was efficient in mitigating the adverse effects caused by water deficit on gas exchange and tomato fruit production under field conditions.

## Figures and Tables

**Figure 1 plants-11-01775-f001:**
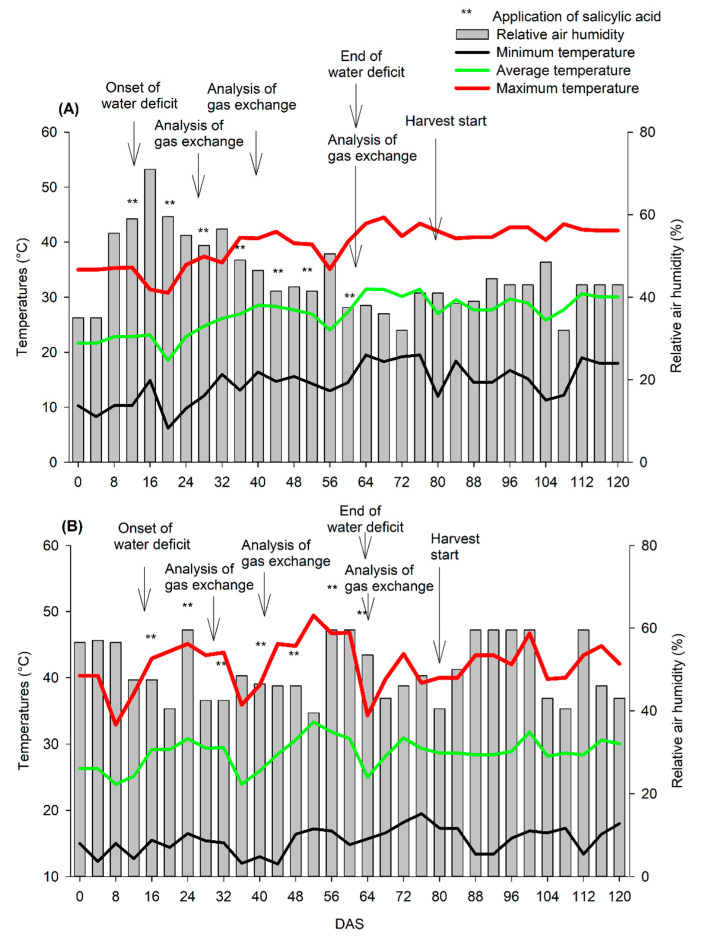
Average, maximum, and minimum temperatures (°C); Relative air humidity (%) inside the greenhouse during 2019 (**A**) and 2020 (**B**) cycles. DAS—Days after sowing.

**Figure 2 plants-11-01775-f002:**
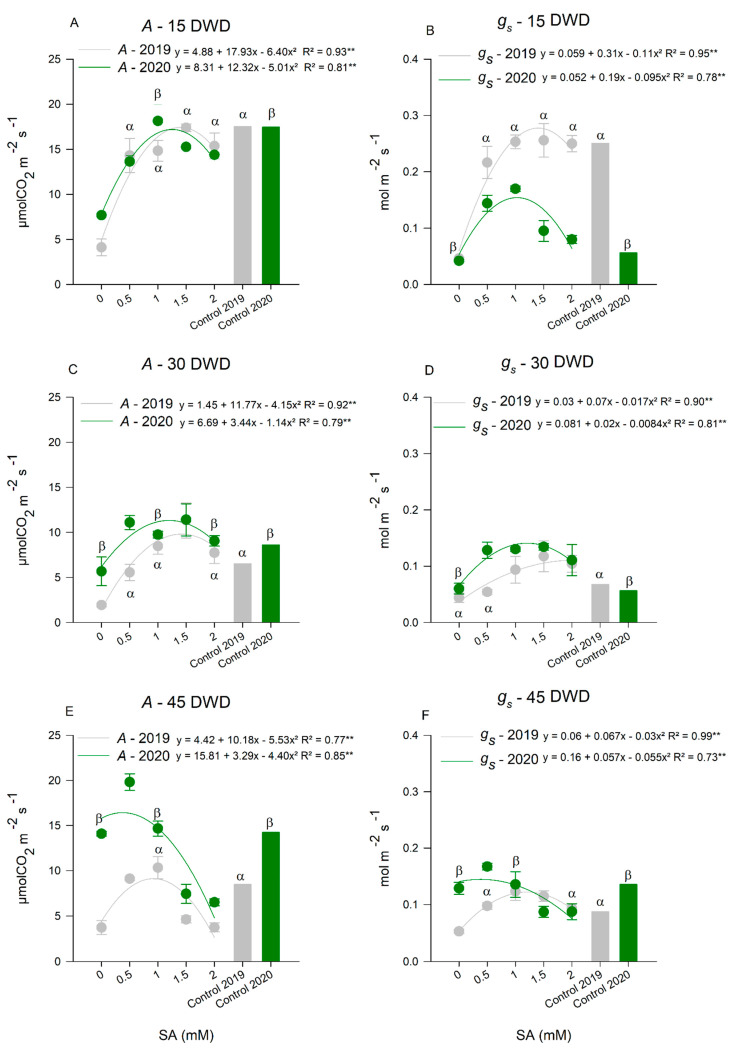
CO_2_ assimilation rate (**A**) at 15 (**A**), 30 (**C**), and 45 (**E**) days of water deficit (DWD); stomatal conductance (g_s_) at 15 (**B**), 30 (**D**), and 45 (**F**) DWD in tomato plants subjected to application of salicylic acid (SA) doses in two consecutive years, 2019 and 2020. ** Highly Significant at 5% probability. ^α^ Indicates an equal and significant effect between treatments by Dunnett’s Test at 5% probability in the 2019 cycle and ^β^ in the 2020 cycle. The bars show the standard deviation. *n* = 4 (number of replicates).

**Figure 3 plants-11-01775-f003:**
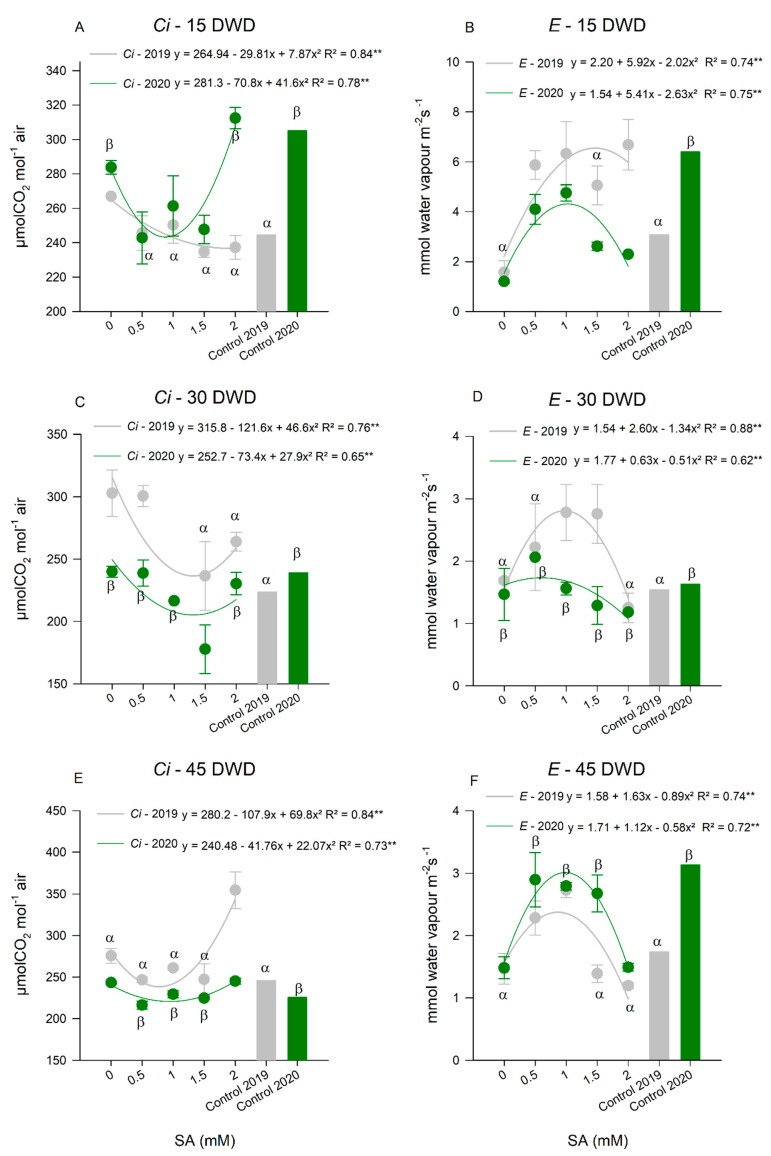
Internal concentration of CO_2_ (Ci) at 15 (**A**), 30 (**C**), and 45 (**E**) days of water deficit (DWD); transpiration (**E**) at 15 (**B**), 30 (**D**), and 45 (**F**) DWD of tomato plants subjected to application of salicylic acid (SA) doses in two consecutive years 2019 and 2020. ** Highly Significant at 5% probability; ^α^ Indicates that there was an equal and significant effect between treatments by the Dunnett’s Test at 5% probability in the 2019 cycle and ^β^ in the 2020 cycle. The bars show the standard deviation. *n* = 4 (number of replicates).

**Figure 4 plants-11-01775-f004:**
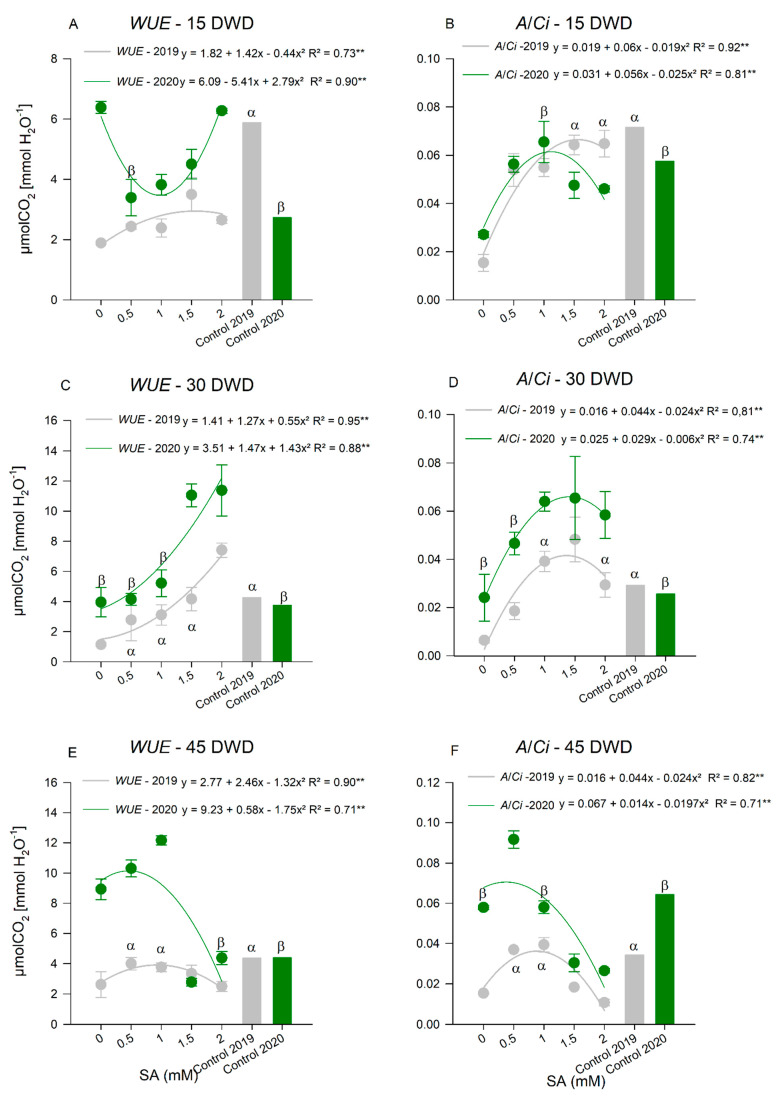
Water use efficiency (*WUE*) at 15 (**A**), 30 (**C**), and 45 (**E**) days of water deficit (DWD); carboxylation efficiency (*A/Ci*) at 15 (**B**), 30 (**D**), and 45 (**F**) DWD of tomato plants submitted to the application of salicylic acid (SA) doses in two consecutive years (2019 and 2020). ** Highly Significant at 5% probability; ^α^ Indicates that there was an equal and significant effect between treatments by Dunnett’s Test at 5% probability in the 2019 cycle and ^β^ in the 2020 cycle. The bars show the standard deviation. *n* = 4 (number of replicates).

**Figure 5 plants-11-01775-f005:**
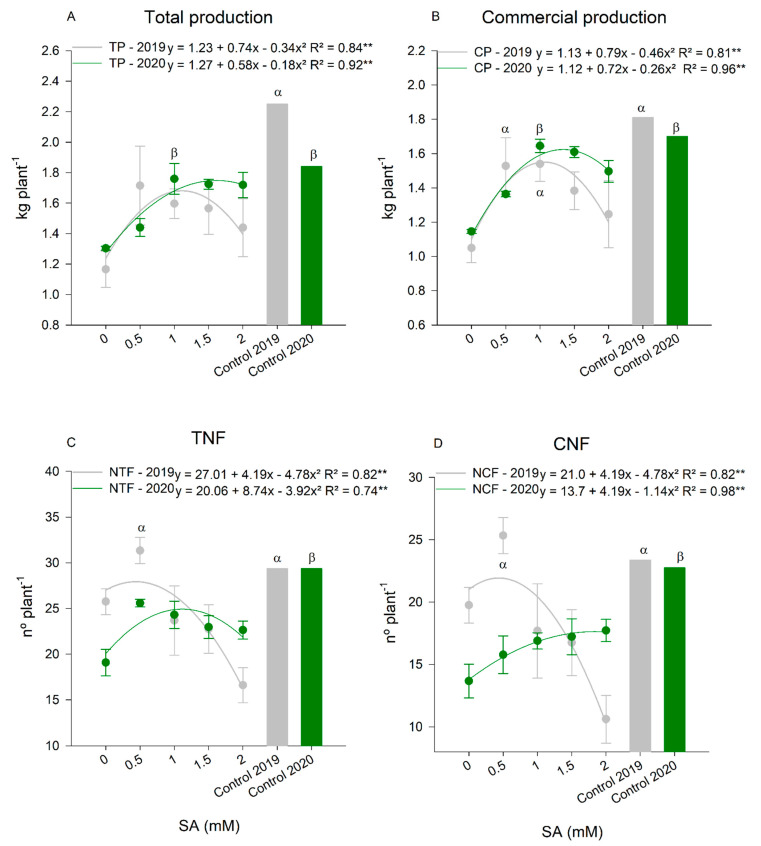
Total production (**A**), commercial production (**B**), total number of fruits (TNF) (**C**), and commercial number of fruits (CNF) (**D**) of tomato plants submitted to the application of doses of salicylic acid (SA) and water deficit in two consecutive years (2019 and 2020). ** Highly Significant at 5% probability; ^α^ Indicates that there was an equal and significant effect between treatments by the Dunnett’s Test at 5% probability in the 2019 cycle and ^β^ in the 2020 cycle. The bars show the standard deviation. *n* = 4 (number of replicates).

**Figure 6 plants-11-01775-f006:**
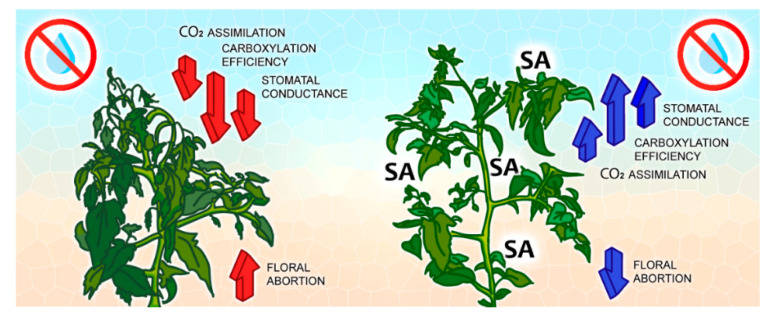
Graphical abstract of Foliar application of salicylic acid to mitigate water stress in tomato.

## Data Availability

All data included in the main text.

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
