# Peer review of "Foliar Application of Salicylic Acid to Mitigate Water Stress in Tomato"

_plants, 2022, doi:10.3390/plants11131775_

Round 1

Reviewer 1 Report

The paper aims to provide comprehensive findings on foliar application of salicylic acid for mitigating the adverse effects of water deficit in tomato. In this regard, the authors have tried to measure physiological parameters related to photosynthetic efficiency and quantified fruit production in plants treated with salicylic acid. This research is indeed important for improving the production of tomato crop in drought affected areas.

The manuscript is well-structured. However, it needs some clarifications and further improvement in the write up.

  1. The title of the manuscript is not explicit, needs rewording.
  2. The terms continuous water stress as a treatment and commercial production is not clear because the experiments was set up in pots on limited number of plants.
  3. The Material and Methods section needs further clarity about SA treatments and water regimes. It appears that SA was applied only to plants grown under restricted water supply not under well-watered conditions.

The manuscript has been reviewed and edited using ‘PDF’. Additionally, specific comments have been added in the text where appropriate. The edited version is attached here for corrections.

Author Response

Dear Reviewer,

Thanks for the contributions in the quality of the manuscript.

We accept all corrections, change the title, add the graphic abstract and change some English terms.

About the term "grape tomato", this is a type of tomato, smaller, sweeter and with smaller areas of cultivation than the Italian one (variety used in this paper), I leave here the link of the article that deals with this work and use the term grape tomato: Chakma, R.; Biswas, A.; Saekong, P.; Ullah, H.; Datta, A. Foliar application and seed priming of salicylic acid affect growth, fruit yield, and quality of grape tomato under drought stress. Scientia Horticulturae, 2021, 280, 109904. https://doi.org/10.1016/j.scienta.2021.109904.

All text changes are highlighted. 

Best regards

Reviewer 2 Report

The manuscript is very interesting, addressing a novel piece of information. I have no hesitation to recommend it for publication in this reputed journal. The philosophy of this work is very good, and applicable in basic and applied plant sciences. The structure and content of the paper are understandable.

Arguments need clearer and tighter presentation and more developed discussion about the impact of drought stress on carbon metabolism. Drought stress decreased the photosynthetic parameters of tomatoes and the authors tested in low water conditions the use of SA.  The authors characterize in detail the potential functional cooperative role of salicylic acid in mitigating the plant environmental stress with a special emphasis on drought. The authors successfully tried to conceptualize the knowledge related to SA and the performance of plants.

Results are applicable in the next applied research and practice and can provoke new experimental activities. The paper can also provide a basis for a better understanding of the impacts and mitigation mechanisms of climate change on crop production. The paper brings new original aspects and the novelty of the paper is OK. Authors confirmed the positive effect of foliar application of SA is efficient in mitigating the deleterious effects of water deficit in tomato plants regarding the gas exchange and fruit production.

To improve readership and the impact of this work, I would like to suggest one schematic diagram in the discussion section.

The paper brings new aspects and novelties. Some arguments need clearer and tighter presentation, more understandable for a large spectrum of plant biologists.

The recent advancement in plant drought tolerance should be better cited. I would like to underline the importance of glycine betaine under drought and other abiotic stresses. The discussion could be improved to make the manuscript easy to follow, and more attractive and critical.

I would have expected a discussion about water stress effects on plant regulation mechanisms. Please, include the missing information (research gaps).

Read/ use papers:

- Li, D., Wang, M., Zhang, T., et al.: Glycinebetaine mitigated the photoinhibition of photosystem II at high temperature in transgenic tomato plants. Photosynth Res (2021). https://doi.org/10.1007/s11120-020-00810-2

- Samui I., Skalicky M., Sarkar S., et al.: Productivity, nutritional quality and water use efficiency of tomato (Solanum Lycopersicum L.) are influenced by drip irrigation and straw mulch in the coastal saline ecosystem of Ganges Delta, India. Sustainability 2020, 12, 6779; doi:10.3390/su12176779

- Li D, Zhang T, Wang M, Liu Y, et al.: Genetic Engineering of the Biosynthesis of Glycine Betaine Modulates Phosphate Homeostasis by Regulating Phosphate Acquisition in Tomato. Front. Plant Sci. 2019, 9:1995. doi: 10.3389/fpls.2018.01995

- Sohag A.A.M., Tahjib-Ul-Arif M., et al.: Exogenous salicylic acid and hydrogen peroxide attenuate drought stress in rice. Plant Soil Environ., 2020, 66: 7–13 https://doi.org/10.17221/472/2019-PSE

I recommend that minor revisions are done to the article before publication. The results can promote future research.

Author Response

Dear Reviewer,

Thanks for the contributions in the quality of the manuscript.

We accept all corrections, change the title, add the graphic and change some English terms.

We added suggested pappers, which made the discussion more robust and clear.

All changes are highlighted.

Best regards
